# Potential of *Kalanchoe pinnata* as a Cancer Treatment Adjuvant and an Epigenetic Regulator

**DOI:** 10.3390/molecules27196425

**Published:** 2022-09-29

**Authors:** Marta Elena Hernández-Caballero, José Alfredo Sierra-Ramírez, Ricardo Villalobos-Valencia, Emmanuel Seseña-Méndez

**Affiliations:** 1Facultad de Medicina, Benemérita Universidad Autónoma de Puebla, Biomedicina, 13 sur 2702 Col. Volcanes, Puebla C.P. 72410, Mexico; 2Sección de Estudios de Posgrado e Investigación, Escuela Superior de Medicina, Instituto Politécnico Nacional, Salvador Díaz Mirón Esq. Plan de San Luis S/N, Miguel Hidalgo, Casco de Santo Tomas, Mexico City 11340, Mexico; 3UMAE Hospital de Oncología, Centro Médico Nacional Siglo XXI, Instituto Mexicano del Seguro Social, Av. Cuauhtémoc 330, Col. Doctores, Cd México C.P. 06725, Mexico

**Keywords:** *Kalanchoe pinnata*, phytochemicals, antitumor activity, adjuvant agent

## Abstract

Cancer is a global public health problem that is related to different environmental and lifestyle factors. Although the combination of screening, prevention, and treatment of cancer has resulted in increased patient survival, conventional treatments sometimes have therapeutic limitations such as resistance to drugs or severe side effects. Oriental culture includes herbal medicine as a complementary therapy in combination with chemotherapy or radiotherapy. This study aimed to identify the bioactive ingredients in *Kalanchoe pinnata*, a succulent herb with ethnomedical applications for several diseases, including cancer, and reveal its anticancer mechanisms through a molecular approach. The herb contains gallic acid, caffeic acid, coumaric acid, quercetin, quercitrin, isorhamnetin, kaempferol, bersaldegenin, bryophyllin a, bryophyllin c, bryophynol, bryophyllol and bryophollone, stigmasterol, campesterol, and other elements. Its phytochemicals participate in the regulation of proliferation, apoptosis, cell migration, angiogenesis, metastasis, oxidative stress, and autophagy. They have the potential to act as epigenetic drugs by reverting the acquired epigenetic changes associated with tumor resistance to therapy—such as the promoter methylation of suppressor genes, inhibition of DNMT1 and DNMT3b activity, and HDAC regulation—through methylation, thereby regulating the expression of genes involved in the PI3K/Akt/mTOR, Nrf2/Keap1, MEK/ERK, and Wnt/β-catenin pathways. All of the data support the use of *K. pinnata* as an adjuvant in cancer treatment.

## 1. Introduction

Cancer is a genetic disease that displays a variety of molecular alterations in the genome of somatic cells [1]. Cancer cells’ traits, such as sustained proliferation, resistance to cell death, angiogenesis capacity, invasion, evasion of immune surveillance, and metastasis, could be the targets of bioactive compounds of medicinal plants. These alterations include small or large structural variations due to epigenetic changes characterized by the addition or removal of chemical groups from/to DNA, histones [2], or RNA [3]. As such, inherited or sporadic mutations cannot only activate oncogenes or deactivate tumor suppressor genes, but may even lead to reversible chemical modifications such as the methylation of DNA, histones, or RNA, which can affect gene expression without changing DNA sequences. Plants from the genus Kalanchoe (Fam: Crassulaceae) have a global distribution in warm climates, where they are used as ornamental plants. Some of the 200 Kalanchoe species are known for their curative uses in different diseases, including cancer. Natural remedies are used on a large scale worldwide, and herbal extracts are obtained from a great variety of plants. At the level of chemical composition, Kalanchoe species include flavonoids, bufadienolides, fatty acids, triterpenoids, alkaloids, phenolic acids, saponins, tannins, glycosides, and kalanchosides [4,5,6]; many of these constituents have remarkable anticancer potential. Numerous studies have evaluated the anticarcinogenic efficacy of these natural bioactive molecules. For example, quercetin has been reported to inhibit the proliferation of human breast cancer cells [7]. Bufadienolides have been shown to have antiangiogenic activity, cause inhibition of cell growth and proliferation, and induce cell death [8]. Antiproliferative activity has also been reported for triterpenes [9] and kalanchosides [5]. Important issues in cancer treatment are drug-related cumulative toxicity and chemoresistance. Hence, combination therapy offers good results due to the use of different targets and reduced adverse effects. Alternative therapy options may include the use of phytochemicals. For example, Lin et al. [10] suggested that the use of natural compounds could offer benefits such as sensitization of tumor cells to drugs, decreased drug efflux from tumor cells, and promotion of repair in normal cells, keeping in mind the potential for herbal–drug interactions. Recent research has also considered the anticancer efficacy of phytochemicals in epigenetic regulation pathways.

This work reviews the features of *Kalanchoe pinnata* (Figure 1) and its phytochemicals, with a focus on the mechanisms underlying its chemopreventive and therapeutic properties.

## 2. Phytochemical Constituents of *Kalanchoe pinnata*

*Kalanchoe pinnata* (Lam) Pers. (syn. *Bryophyllum pinnatum* Lam.), belonging to the genus Kalanchoe, is a succulent plant cultivated in gardens [11], which can be found as a herb or shrub. The plant has opposite, simple, and compound leaves, with a red to dark purple crenate margin. It reproduces from seeds and vegetatively through leaves [12], presenting clusters of reddish-purple pendulous flowers [13]. *K. pinnata* is used in traditional medicine to treat different diseases including cancer [6]. Mora-Pérez and Hernández-Medel [14] analyzed the ingredients of *K. pinnata* and found alkaloids and sterols in methanolic root extract, and terpenes, sterols, flavonoids, chlorides, nitrates, and potassium in methanolic stem extract. The leaves of this plant contain phenols (gallic acid), flavonoids (quercetin) (in methanol extract), lycopenes, and β-carotenes (in petroleum ether) [15], as well as tannins and alkaloids [16]. Jaiswal et al. [17] found that the phenolic content of leaves was 28.4 ± 2 µg mg^−1^ and suggested that this was responsible for their antioxidant capacity. El Abdellaoui et al. [18] detected three phenolic acids (gallic, caffeic, and coumaric acids), three flavanol glycosides (quercetin, isorhamnetin, and kaempferol), 4′,5-dihydroxy-3′,8-dimethoxyflavone 7-O-β-D-glucopyranoside, and quercitrin [19]. Bufadienolides such as bersaldegenin acetate-2, -3, -4, -5, bryophyllin a, bryophyllin c, and bersaldegenin-1,3,5-orthoacetate [20] have also been found. Reported phenanthrene derivatives include Ψ-taraxasterol and 18-α-oleanane, and other ingredients include sterols; bryophynol, bryophyllol, and bryophollone [21]; stigmasterol [22]; campesterol, 24-epiclerosterol, (24R)-5α-stigmasta-7,25-dien-3β-ol, 5 α-stigmast-24-en-3β-ol, and 25-methyl-5α-ergost-24(28)-en-3β-ol [23] (Figure 2). The leaves of *K. pinnata* are rich in ascorbic acid (vitamin C) and contain riboflavin, thiamine, niacin, magnesium, calcium, potassium, phosphorus, sodium, and microelements such as iron and zinc [16]. *K. pinnata* flowers contain a higher concentration of glycosides [24], similar to other *Kalanchoe* species.

## 3. Activity of *Kalanchoe pinnata* in Cancer

Several plant components have been identified as sources of anticancer therapeutics. However, those components should be investigated in clinical trials to confirm their pharmacokinetic effects. Members of the Kalanchoe genus (Crassulaceae) have remedial properties for a wide range of diseases such as gastric ulcers, urolithiasis, bacterial, viral, and parasitic infections, skin diseases, cold, memory improvement, or even improvement in sleep quality when undergoing cancer treatment. Based on ethnobotanical evidence, the Kalanchoe genus has been evaluated on various cancer cell lines (Table 1). The phytochemicals present in K. pinnata have been analyzed in a variety of studies on cancer.

Plant extracts by themselves do not produce a significant effect against cancer, but they may enhance therapeutic efficacy when combined with chemotherapeutic drugs. The most studied natural compounds are polyphenols, which are widely distributed in plant tissues. These compounds are secondary metabolites produced for protection against bacteria, fungi, and insects. Natural phenols more studied for their properties are curcumin, epigallocatechin-3-gallate, resveratrol, quercetin, and myricetin [25], of these compounds, quercetin is found in *K. pinnata*.

### 3.1. Phenolic Acids

#### 3.1.1. Gallic Acid

Gallic acid (3,4,5-trihydroxybenzoic acid, GA) is a phenolic acid belonging to the tannin family. It is found in a variety of fruits and vegetables, either in a free form or in the form of tannins [41]. Acute oral administration of GA at a concentration of 500 mg/kg body weight is nontoxic, and subacute administration at a dose of 1000 mg/kg body weight has not shown toxic side effects in mice [42]. Reproductive toxicity in rats was also evaluated, and no evidence of toxicity at 430 mg/kg was found [43]. In addition to the antioxidant and anti-inflammatory effects, several reports have supported the anticancer function of GA. GA affects cancer cells by regulating angiogenesis, proliferation, and apoptosis. Mirvish et al. [44] reported a strong inhibition of lung adenomas in mice by GA added to the food. Inoue et al. [45] proposed that GA triggers apoptosis in HL-60RG cells through the formation of reactive oxygen species (ROS) and calmodulin activation. ROS are involved in the regulation of many processes by up- or downregulation of critical protein kinase activities and are the cause of intracellular oxidative stress. Prostate, stomach, liver, skin, leukemia, lymphoma, colon, breast, cervical, esophageal, and endometrial cancer cell lines have been investigated for GA activity as reviewed elsewhere [46]. Studies on cancer cell lines have found not only apoptosis induction by the release of cytochrome c from mitochondria and the activation of caspases but also ferroptotic cell death and necroptotic pathways [47]. Tang and Cheung [47] identified enrichment of AMPK, TNF, and mTOR signaling pathways in GA-induced HeLa cells. Using an in vivo mouse model, Ko et al. [48] found that GA regulated the PI3K/Akt pathway in lung cancer cell line A549-derived tumors. When comparing the effects of GA and cisplatin, the authors found that GA inhibited cell proliferation and colony formation by inhibiting the PI3K/Akt pathway and upregulating p53. According to Isuzugawa et al. [49], GA induces apoptosis through the generation of hydrogen peroxide. Moreover, they found a great difference in catalase levels between hepatocellular carcinoma cells and normal hepatocytes, suggesting that this difference is key to the GA response. Indeed, catalase is important to hydrogen peroxide provision, which is why GA is more effective against tumor cells than against nontumor cells. Due to its low cytotoxic effects, GA is a promising chemopreventive agent. The use of nanoparticle-based drug delivery systems has achieved many advances in recent years. For example, hyaluronic acid (HA) is an attractive biopolymer as it has the advantage of affinity for CD44—a receptor highly expressed on various cancer cells [50]. Nanoparticles of HA conjugated with polyethylene glycol-distearoyl-phosphoethanolamine co-loaded with doxorubicin and GA have been tested in doxorubicin-resistant human HL promyelocytic leukemia cells, showing good results in inhibiting tumor growth [51]. DNA and RNA methylation and histone modifications are other key elements in the etiology of cancer. Phytochemicals can modulate the gene expression of writer and reader proteins, which are involved in the addition and removal of epigenetic marks.

#### 3.1.2. Caffeic Acid

Unlike GA, caffeic acid (CA), or hydroxycinnamic acid [(E)-3-(3,4-dihrydroxyphenyl) prop-2-enoic acid], is an antiapoptotic polyphenol. It is almost fully absorbed in the small intestine and is nontoxic even at daily doses of 0.5–1 g [52]. Rosendahl et al. [53] reported that CA was associated with a moderate reduction in estrogen receptor-α (ER)^+^ and ER^−^ human breast cancer cell proliferation via ER and insulin-like growth factor-I receptor (IGFIR). Neuroblastoma SH-SY5Y cells are protected from cyclophosphamide toxicity by CA; specifically, cell death is suppressed through a decrease in lipid peroxidation levels [54]. The concomitant action of CA and cisplatin in ovarian carcinoma cell line A2780 enhances cisplatin cytotoxicity, but the use of CA before cisplatin treatment causes resistance through phase II induction of glutathione S-transferase (GST) and glutathione reductase (GR) by induction of the Nrf2/Keap1 pathway [55]. Additionally, some studies have found that CA can inhibit the migration of cancer cells [56,57].

#### 3.1.3. P-coumaric Acid

*p*-coumaric acid (*p*-CA) is a hydroxycinnamic acid derivative of cinnamic acid. It has been shown that *p*-CA protects against oxidative stress and DNA and chromosome breakage in HT-29 cells [58]. *p*-CA has an antiproliferative effect on Caco-2 cells [59]. In an induced model of colon carcinogenesis, it was shown that *p*-CA reduced preneoplastic changes and stimulated apoptosis [60]. Gastaldello et al. [61] reported a reduction in angiogenesis and the presence of intratumoral neutrophils in rats inoculated with B16F10 melanoma cells.

### 3.2. Flavonoids

#### 3.2.1. Quercetin

Quercetin (3,5,7,3′,4′-pentahydroxyflavone, Qu) is an important bioflavonoid found in vegetables in its glycoside form. Like phenolic acids, it is absorbed in the small intestine. Oral administration of 1 g/day is safe [62]. Wätjen et al. [63] suggested a dose of 10–25 µmol/L Qu acted against H_2_O_2_-induced cytotoxicity, DNA strand breaks, and caspase-3 activation; the authors also suggested that Qu induced cytotoxicity at concentrations between 50 and 250 µmol/L. Qu protects cells from oxidative stress, thereby reducing the levels of ROS. Many studies have shown that Qu represses the proliferation of different cancer cell lines. In endometrial cells, 100 µM Qu-inhibited DNA synthesis and cell growth by suppressing cyclin D and EGF [64]. In addition, Qu exhibited an antiangiogenic effect by suppressing VEGF-induced phosphorylation of VEGF receptor 2 and signaling pathways that involve protein kinases AKT, mTOR, and ribosomal protein S6 in human umbilical vein endothelial cells (HUVECs) and prostate tumor cells [65]. Qu can be used as an antimetastatic agent as it is structurally homologous to LY 294002 (LY), the commercially available inhibitor of PI3K; both agents are effective in suppressing the Akt/PKB pathway through PIP3 regulation [66]. In combination with cisplatin, 5 µM Qu sensitizes osteosarcoma 143B cells to cisplatin, while in concentrations ≥ 10 µM, Qu inhibits the proliferation of these cells [67]. While Qu inhibits cell proliferation and invasion and increases apoptosis and PTEN expression, it downregulates the expression of p-Akt in MCF-7 cells [68]. Using mouse breast tumor cells (EMT6), Liu et al. [69] reported that Qu plus cisplatin synergistically inhibited tumor growth, reduced renal toxicity induced by cisplatin, and increased the cytotoxicity of cisplatin. The co-delivery of Qu and paclitaxel reduced multidrug resistance (MDR) in breast cancer cells with multidrug-resistance MDA-MB-231/MDR1 and activated mitochondria-dependent apoptosis [70]. In glioblastoma multiforme, Qu combined with temozolomide, using HA as a nanocarrier, showed a synergistic effect on temozolomide efficacy. The CD44 receptor was the preferred target, enhancing the anti-inflammatory mechanism and decreasing interleukin secretion [71]. As for drugs used to treat advanced carcinomas, Qu enhanced the efficacy of gemcitabine and doxorubicin, and it reduced resistance to chemotherapy by S-phase arrest, apoptosis induction, and increased p53 expression [72].

#### 3.2.2. Quercitrin

Quercitrin (quercetin 3-O-α-L-rhamnopyranoside, Qi) is a glycoside of Qu [73]. It acts as an antioxidant by scavenging free radicals [74]. Similar to Qu, Qi has antiproliferative and apoptotic effects on different cell lines, such as colorectal and lung cancer cell lines, where it has been shown to increase caspase-3 activity and decrease mitochondrial membrane potential [75,76]. Tumor initiation, promotion, and progression are prompted by different transcription factors such as AP-1 and NF-κB; these are activated by mitogen-activated protein kinases (MAPKs), c-jun NH-2 terminal kinase, or extracellular signal-regulated kinases. It has been suggested that Qi inhibits NF-κB and AP-1-MAPKs signaling and induces the activity of phase II detoxifying enzymes such as GST and NQO1 in JB6 and JB6 P+ mouse epidermal cell lines, which gives Qi potential as an anticancer agent [77].

#### 3.2.3. Isorhamnetin

Isorhamnetin (isorhamnetin-3-O-methylquercetin, Iso) is a flavonoid metabolite of quercetin. It has been found that Iso participates in the PI3K/AKT pathway, thereby suppressing proliferation and metastasis and triggering G2/M phase arrest in gallbladder cancer [78]. Inhibition of Src activity and inhibition of β-catenin nuclear translocation by induction of C-terminal Src kinase expression were the results of Iso incorporation into the diet of a mouse model of colorectal cancer and in HT-29 colon cancer cells [79]. Recently, it has been found that Iso promotes apoptosis of gastric cancer MKN-45 cells in a hypoxic environment by inhibiting PI3K/AKT/mTOR-mediated adaptive autophagy [80]. In breast cancer cell lines, Iso inhibits the AKT/mTOR and MEK/ERK pathways [81]. Considering that the PI3K/AKT/mTOR pathway has been associated with tamoxifen resistance, Iso has an antitumor action [82]. In gastric cancer, the combination of Iso with capecitabine enhanced the apoptotic effect of capecitabine by avoiding NF-κB activation in MKN-45 cells [83]. Zhang et al. [84] found that non-small cell lung cancer cell line A-549 pretreated with Iso had a better response to radiotherapy, probably thanks to increased expression of IL-3 and inhibited activation of NF-κB.

#### 3.2.4. Kaempferol

Similarly to Iso, kaempferol 3,5,7-trihydroxy-2-(4-hydroxyphenyl, Kae)-4H-1-benzopyran-4-one [85] belongs to the flavanols. Kae has been reported to induce apoptosis of breast cancer cell line MCF-7 intrinsically via apoptotic downregulation of Bcl2, upregulation of Bax expression, and cleavage of PARP [86]. In ovarian cancer cells A2780/CP70, Kae stimulated the extrinsic apoptosis pathway through death receptors/FADD/caspase-8; moreover, Kae induced G2/M cell cycle arrest via two pathways, Chk2/Cdc25C/Cdc2 and Chk2/p21/Cdc2. Doxorubicin induces cellular apoptosis, but it has significant side effects, and its therapeutic effect is limited by acquired drug resistance. In combination with Kae, doxorubicin inhibits viability, colony formation, cell cycle progression, DNA damage response, and mitochondrial function in liver cancer cells. The combined treatment has a stronger inhibitory effect on the PI3K/mTOR/MMP pathway than either alone [87]. In the same way, Kae plus erlotinib is more efficient at inhibiting PI3K/AKT in pancreatic cancer cells PANC-1 and BxPC-3 targeting EGFR and promoting apoptosis [88]. In vitro data reveal that the actions of many flavonoids converge on the PI3k/Akt/mTOR pathway, making them good antitumor candidates. Flavonoids can inhibit the efflux function of ATP-binding cassette (ABC) transporters, such as ABCB1 and ABCG2, thereby increasing the bioavailability of poorly available drugs; yet, they can potentiate the toxicity of some ABC transporter substrates, such as chemotherapeutic drugs [89].

### 3.3. Sterols

#### 3.3.1. Bufadienolides

Bufadienolides are made up of polyhydroxy C-24 steroids and their glycosides, which have an α-pyrone ring at the C-17β position [90]. Their name comes from the genus *Bufo*, given that toad skin contains poison glands with these compounds. Bufadienolides show insecticidal, antimicrobial, anti-inflammatory, cardiotonic, anesthetic, sedative, and antitumoral activities [91]. Bufadienolides are involved in diverse processes such as apoptosis, autophagy, cell cycle arrest, angiogenesis, epithelial–mesenchymal transition, and metastasis [92].

Bersaldegenin-1,3,5-orthoacetate plays a role in cell cycle arrest and caspase-independent cell death in HeLa cells [8]. Xiuzhen et al. [93] reported its potent cytotoxicity in human lung carcinoma A-59 and colon HTC-8 cell lines, which could be attributed to the presence of bersaldegenin-1,3,5-orthoacetate in chloroform extract from a whole fresh plant of *Bryophyllum pinnatum* collected in Taipei, Taiwan.

Bryophyllin a is a bufadienolide-1,3,5-orthoacetate with potent cytotoxicity against KB, A-549, and HCT-8 cells [94]. Bryophyllin c [95] and a have been reported to inhibit tetradecanoylphorbol acetate-induced Epstein Barr virus activation in Raji cells [28]. Other bufadienolides present in *K. pinnata*, such as bryophynol, bryophyllol, and bryophollone, or the sterol 24-epiclerosterol, have not been examined in terms of their antitumor activity.

#### 3.3.2. Stigmasterol

Stigmasterol (3beta-hydroxystigmastane with double bonds at the 5,6- and 22–23 positions, St) is a phytosterol [85]. Li et al. [96] demonstrated that St-inhibited A278 and SKOV3 ovarian cancer cell proliferation and migration, reduced mRNA expression of PI3K/Akt, and increased the expression of PTEN. In SGC-7901 and MGC-803 gastric cancer cells, St-inhibited viability and proliferation and induced autophagy by blocking the Akt/mTOR pathway, with autophagy playing a cryoprotective role against apoptosis induced by St [97]. When Nrf2, a transcription factor involved in the induction of phase II enzymes and endogenous antioxidants, is overexpressed, chemoresistance is observed in many cancers [98]. Liao et al. [99] analyzed the role of St in the chemoresistance of the endometrial cancer cell line Ishikawa overexpressing Nrf2; they found that the overexpression of Nrf2 reduced sensitivity to cisplatin and that St-sensitized cells to chemotherapy by suppressing Nrf2 expression. In another study, St-reduced the activity of P-glycoprotein efflux, a membrane transporter, in drug-resistant human leukemia CEM/ADR5000 cells at nontoxic concentrations, and it increased the sensitivity to doxorubicin in drug-resistant cancer colon cells Caco-2, thereby reestablishing doxorubicin cytotoxicity [100]. Furthermore, St was successfully used in a phyto-liposome with hyaluronic acid-modified PEGylated doxorubicin in breast cancer MDA-MB-231 cells, especially in CD44-overexpressing cells [101].

#### 3.3.3. Campesterol

Campesterol (Cam) is a phytosterol, a 3-beta-sterol, and a 3beta-hydroxy-Delta(5)-steroid [85]. The analysis of its effect on ovarian cancer ES2 and OV90 cells provided information on its participation in the activation of cell death signals, reduced mitochondrial membrane function, and apoptosis. Similar to other sterols, Cam affects the PI3K/MAPK pathway and inhibits cell cycle proliferation by targeting PCNA. Additionally, Cam enhances the effects of cisplatin and paclitaxel [102].

## 4. Participation of *K. pinnata* in Epigenetic Regulation

Gene expression is regulated by diverse factors, including epigenetic regulation; major epigenetic mechanisms include DNA methylation, histone modifications, chromatin remodeling, microRNA, and RNA methylation (Figure 3). Research has accumulated on several of these mechanisms, while epitranscriptomics is an area that is currently being intensively investigated. Weng et al. [103] demonstrated that GA inhibited nuclear and cytoplasmic DNA methyltransferase DNMT1 in lung cancer cell line H1299 after seven days of treatment; they also found that GA reduced phosphorylation of DNMT1, thereby allowing for its methylation and destabilization, and leading to degradation. p-CA acts as histone deacetylases (HDAC) inhibitor [104] in breast cancer MCF-7 and cervical cancer HeLa cells; altering histone acetylation leads to increased p21 expression. Qu can also decrease the transcript levels of DNA methyltransferases (DNMTs) DNMT1, 3A, and 3B, along with the enzymatic activity of DNMTs. It has been suggested that Qu binds to the polycomb repressor protein EZH2, thereby suppressing its ability to recruit DNMT. Furthermore, Qu could reduce the expression of class II HDACs, degrade class I HDACs that induce apoptosis, block the cell cycle and angiogenesis, and reduce methylation of the tumor suppressor genes’ promoter in cancer cell lines [105,106]. Treatment of leukemia cell lines HL60 and U937 with Qu-induced demethylation of BCL2L11 and DAPK1 genes, inhibition of HDAC, enrichment of H3ac and H4ac in the promoter of genes involved in apoptosis, and downregulation of DNMT via a STAT3-dependent mechanism; thus, Qu enhanced apoptosis through epigenetic regulation [107]. Kae, similar to Qu, has an epigenetic role, acting as an HDAC inhibitor in hepatoma cell lines HepG2 and Hep3B and colon cancer cell line HCT-116. Kae induces hyperacetylation of histone complex H3 [108]. Lu et al. (2018) found that Kae increased DACT2 expression in colorectal cancer cells HCT116, HT29, and YB5 by DACT2 demethylation through the suppression of DNMT1 and DNMT3b expression, thereby inhibiting the nuclear β-catenin expression and deactivating Wnt/β-catenin pathway proteins [109].

RNA presents numerous modifications but N6-methyladenosine (m6A) is the most abundant (110). m6A is located by a ~900 kDa methylosome complex that contains METTL3, METTL14, WTAP, KIA1429, RBM15, KIAA0853, and HAKAI proteins. The m6A mark is erased by proteins FTO and ALKBH5, and it is decoded by proteins that contain the YTH domain [110]. Du et al. [111] recently published a docking-based high-throughput screening report, in which they found that Qu decreased the m6A level in a dose-dependent manner in MIA PaCa-2 pancreatic cancer cells. Qu inhibited the proliferation in MIA PaCa-2 and Huh7 cells. They found that Qu took the place of METTL3 and formed a stable protein-ligand complex, thereby avoiding the METTL3 function and decreasing the m6A/A ratio. Here, it is important to point out that epigenetic modifications of both DNA and RNA are reversible, unlike mutations, which makes them a promising target in the development of therapies.

Cancer remains an important cause of death worldwide. While new drugs have contributed to improved treatment options, it is important that we find more efficient approaches with lower toxicity for patients. Considering the above information, Qu is one of the most promising phytochemicals present in *K. pinnata*, with well-known pharmacokinetics and a lack of systemic toxicity. The use of herbs from traditional medicine is a promising alternative already used in Oriental culture. In a meta-analysis, Sun et al. [112] summarized the clinical efficacy of Kang-ai injection (an extract from three Chinese herbs) in combination with conventional treatment of hepatocellular carcinoma. Based on the results obtained from its bioactive components, the potential is noted for *K. pinnata* to act as an antioncogenic adjuvant agent. *K. pinnata* may regulate the expression of genes, especially by targeting the PI3K/Akt/mTOR, Nrf2/Keap1, MEK/ERK, and Wnt/β-catenin pathways (Figure 4).

## 5. Pathways Involved in Cancer and Potential Targets to *K. pinnata* Phytochemicals

Cellular metabolism is regulated by various signaling pathways which widely crosstalk. Aberrant regulation of signaling pathways induces disturbance in cellular metabolism and causes diseases. Cancer cells have a complex interplay among different pathways, some components commonly associated with carcinogenesis are Mitogen-Activated Protein Kinase (MAPK) or Extracellular-signal-Regulated Kinase (ERK), Wnt/β-catenin, Phosphatidylinositol 3-kinase (PI3K)/AKT, mammalian Target of Rapamycin (mTOR), and Nuclear factor-E2 p45-related factor (Nrf2).

### 5.1. MEK/ERK

The most studied of the four MAPK pathways include ERK1/2, which is involved in the transmission of mitogenic signals. ERK1/2 acts downstream of Ras, and it involves phosphorylation and sequential activation of kinases. The activation begins with a membrane receptor (receptor tyrosine kinase, G-protein-coupled receptor, or other) which transmits the signal by recruiting adaptor proteins such as Grb2. Recruitment of adaptor proteins induces the activation of Ras, followed by activation of Raf-1, B-Raf, and A-Raf. Upon the activation of Raf proteins, MEK1/2 is phosphorylated, and they finally activate ERK1/2, which are the executors of the upstream signals. Phosphorylation of ERK1/2 on threonine and tyrosine residues allows their nuclear translocation to regulate targets such as Elk1, c-Fos, and c-Jun [113]. The treatment of cell lines with the phytochemicals mentioned above has made it possible to determine that the main effect on the ERK pathway is the phosphorylation of key components such as the ERK1/2 effectors. In HeLa and HTB-35 cells, it was demonstrated that GA reduces the phosphorylation levels of ERK protein suppressing cancer progression [114]. On the other hand, GA can inhibit ERK through the phosphorylation of EGFR, thereby decreasing the up-regulation of metalloproteinase 9 (MMP-9) and reducing the invasive capacity of tumoral cells [115]. CA and *p*-CA also phosphorylate ERK to promote translocation of Nrf2 from the cytoplasm to the nucleus, activating the expression of detoxification enzymes HO-1, GCLC, and GLCM as protection against oxidative damage in HepG2 cells [116]. Qu phosphorylates ERK and c-Jun thus promoting the cleavage of caspase-3 and inducing apoptosis in A549 cells [117]. In MDA-MB-231, HeLa, and SiKa cells, Qu induces changes in ERK1/2 phosphorylation leading to induction of apoptosis, inhibition of cell survival [118,119], and inhibition of colon cancer cell migration and invasion. Qi showed the same effects [120]. Iso suppressed EGF-induced phosphorylation of ERKs, Akt, and p70 by inhibition of MEK1 [121]. Kae also suppresses the phosphorylation of ERK and induces autophagy and apoptosis in ovarian cancer cells [122].

### 5.2. Wnt/β-catenin

Cysteine-rich glycoprotein Wnt is a ligand that can trigger intracellular signal transduction pathways such as the Wnt/β-catenin pathway. Wnt binds to the Frizzled (FZD) receptor and LRP-5/6 co-receptors. This activates Disheveled and in turn, this causes recruitment of the complex (Axin, GSK3 β, APC, CK1) to the receptor. The Wnt-Frizzled-Axin-LRP-5/6 complex sequesters cytosolic GSK3 β rendering it incapable of phosphorylating β-catenin. Accumulated non-phosphorylated β-catenin in the cytosol migrates to the nucleus, there β-catenin functions as a transcriptional co-activator interacting with T cell-specific factor (TCF)/lymphoid enhancer-binding factor (LEF) and binding to the promoters of target genes. Without Wnt signaling, cytoplasmic β-catenin is degraded by the destruction complex (Axin-APC-GSK3 β) [123,124]. Experiments using GA have shown that it suppresses the expression of β-catenin at mRNA and protein levels. Such downregulation suppresses downstream target genes such as VEGF, Oct3/4, surviving, and CCND1 in hepatocellular carcinoma cells.

Furthermore, in a xenograft model, E-cadherin and Bax were up-regulated, and vimentin, MMP9, Bcl2, and Bcl-xl were down-regulated. Authors hypothesized that GA inhibits the epithelial-mesenchymal transition polarization and blocks the potential of metastatic dissemination and induces apoptosis [125]. In human teratocarcinoma cells NT2/D1, Qu inhibits β-catenin translocation, this avoided the expression of genes involved in proliferation, migration, and invasion in diverse cell lines [126,127]. Liu et al. [128] found recently that Qu increases Sirtuin 6, this could reduce transcripts levels of FZD4 and prevent β-catenin cytoplasmic accumulation and translocation.

### 5.3. PI3K/AKT/mTOR

PI3K belongs to a family of lipid kinases that phosphorylate the 3′OH group of phosphatidylinositol, PI3K activation initiates a signal transduction cascade. Akt is a serine-threonine kinase that is directly activated in response to PI3K due to its amino-terminal PH domain which is bound by PIP_3._ Akt is one of the PI3K effectors, its activation results in cell survival, proliferation, growth, and angiogenesis. There are many downstream effectors, such as GSK3 β, FOXO, and MDM2, which in turn are controlled by compensatory pathways. One of the major effectors downstream of Akt is the serine/threonine kinase mTOR which is a key regulator of cell growth [129,130]. GA downregulates protein levels of p-AKT, PI3K, and mTOR affecting cell viability, proliferation, invasion, and angiogenesis in lung cancer, ovarian cancer, and glioma cells [48,131,132]. p-Akt was significantly decreased by CA on colorectal cancer stem cells [133]. In prostate cancer cells, breast cancer cells, and pancreatic cells, Qu binds PI3K suppressing AKT phosphorylation and inducing apoptosis by regulation of the Bax, Bcl2, and p53 expression [119,134]. In the same way, Qi, Sti, and Kae induce apoptosis by inhibition of PI3K phosphorylation [88,135,136], and Iso down-regulated the expression of p-PI3K, p-Akt, and p-mTOR inhibiting cell migration and invasion by matrix metalloproteinases regulation in prostate cancer [137] and skin cancer cells [121].

### 5.4. Nuclear Factor-E2 p45-Related Factor (Nrf2)

Nrf2 belongs to the cap’n’collar subfamily of transcription factors with a basic leucine zipper domain and plays a key role in the cellular stress response against oxidative insults such as electrophiles or cytotoxic agents. Nrf2 persists in an inactivated status, it binds to Kelch ECH associating protein 1 (Keap1), a repressor protein cysteine-rich that promotes Nrf2 degradation by the ubiquitin-proteasome pathway. The oxidative stress causes Nrf2 dissociation and its subsequent translocation into the nucleus to form heterodimers with small Maf proteins and bind to the antioxidant response element (ARE) and induce transcription of phase II enzymes, GST, NQO1, HO-1, GSC, and GCL [138,139]. CA induces Nrf2 translocation to the nucleus and oxidation of the Keap1 protein. CA possesses a dual mechanism of action, acts as a GSTP1 and GSR1 inducer and as their inhibitor in ovarian carcinoma cells A2780 [140], and CA in these cells before cisplatin treatment induces GST and GR expression [55]. Qu and Kae were found could be used as Keap1 inhibitors and Nrf2 activators [141]. Qu increased nuclear localization of Nrf2, enhanced ARE-binding complexes, and caused NQO1 transcription in HepG2 cells. Besides, Qu reduced the level of Keap1 by inhibiting the ubiquitination and destruction of Nrf2 [142] and increased the expression of HO-1 and NOS1 in a neuroblastoma cell line [143].

As we observed, the components of *K. pinnata* converge and affect pathways involved in various processes of cancer development and progression, and they also affect key genes such as p53 and p21. As recently reviewed by Ng et al. [144], there is a large amount of experimental data with plant extracts or fractions, but cancer characteristics and interactions among components in the extracts or in the human body upon administration need more analysis. There is not enough clinical data concerning the application of phytochemicals for cancer treatment, but their potential use as co-treatment is promising.

## 6. Conclusions

This review demonstrated that the use of phytochemicals in combination with oncologic treatments can give promising results in reducing the drug dose. Phytochemicals have been tested in many in vitro studies with cell lines; as such, it is necessary to test them in animal models as possible adjuvants in chemotherapeutic treatment. The perennial herb widely distributed in tropical areas, *K. pinnata*, has components that could repress hallmarks of cancer, such as proliferation, apoptosis, cell migration, angiogenesis, or metastasis, and regulate processes such as oxidative stress or autophagy. They could also act as epigenetic drugs, reverting the acquired epigenetic changes associated with tumor resistance to therapy such as the promoter methylation of suppressor genes, inhibition of DNMTs activity, and HDAC regulation through methylation.

## Figures and Tables

**Figure 1 molecules-27-06425-f001:**
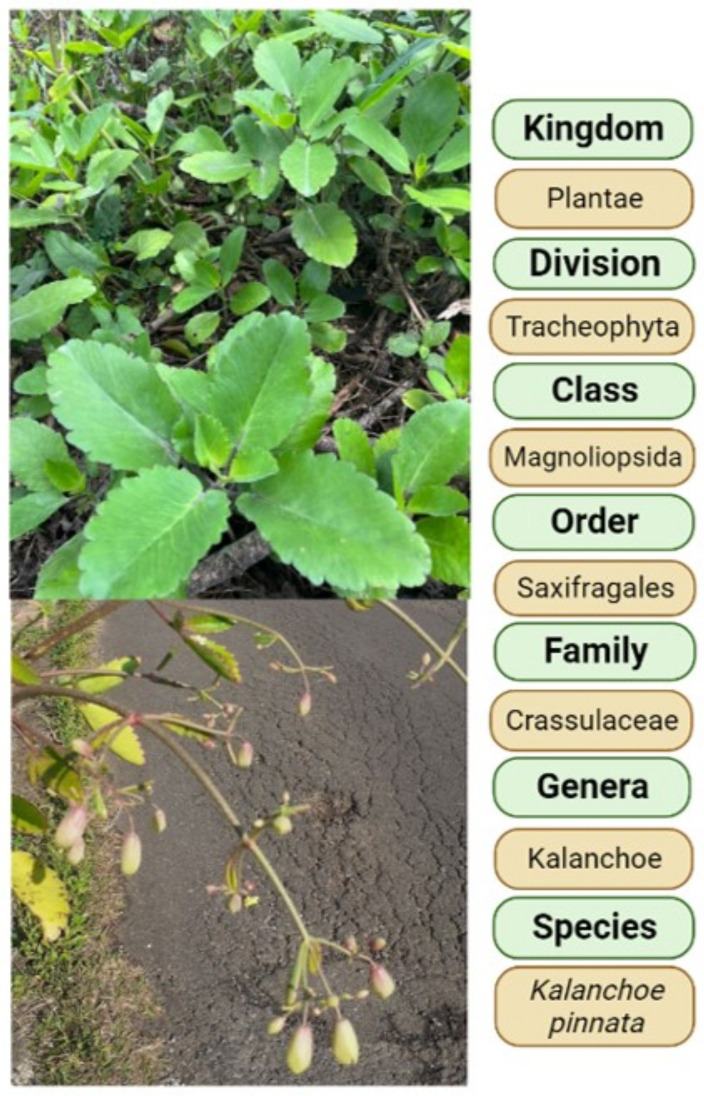
Taxonomic classification of K. pinnata. Source: Photos by authors, image created with BioRender.com.

**Figure 2 molecules-27-06425-f002:**
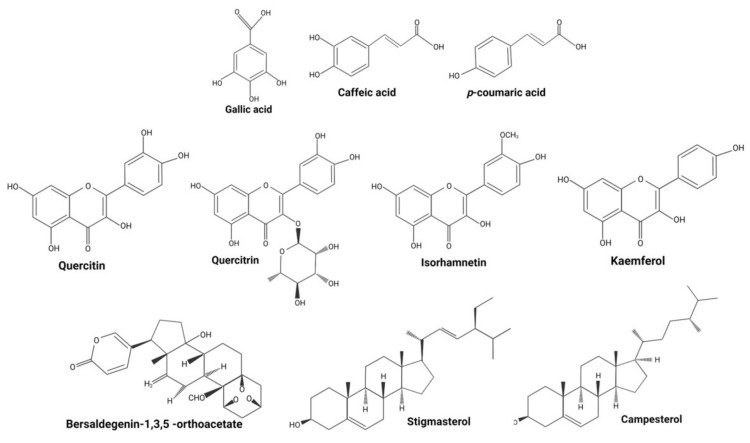
Structures of major phytochemicals reported in *K. pinnata*. Source: Image created with BioRender.com.

**Figure 3 molecules-27-06425-f003:**
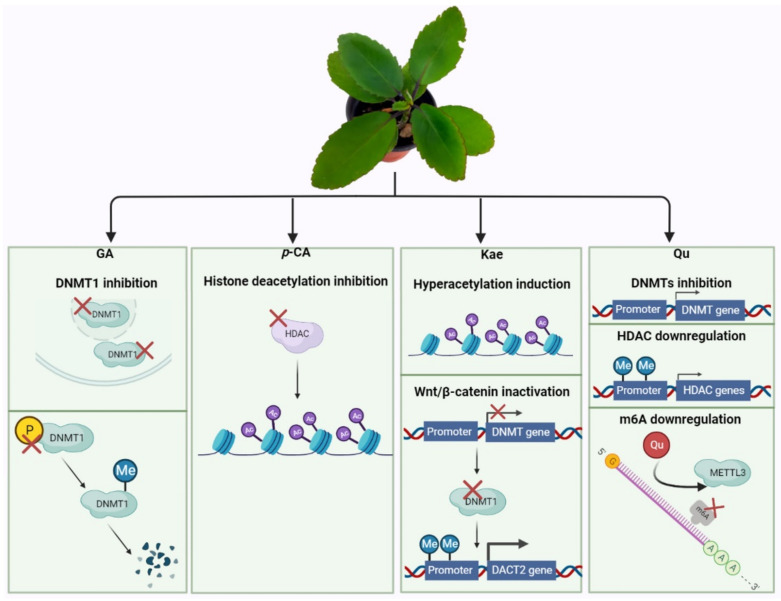
Epigenetic regulation by *K. pinnata* phytochemicals. Some bioactive chemical constituents have a role in epigenetic mechanisms in DNA and RNA. GA: Gallic acid; p-CA: *p*-coumaric acid; Kae: kaempferol; Qu: quercetin; Me: methyl; Ac: acetyl. Source: Image created with BioRender.com.

**Figure 4 molecules-27-06425-f004:**
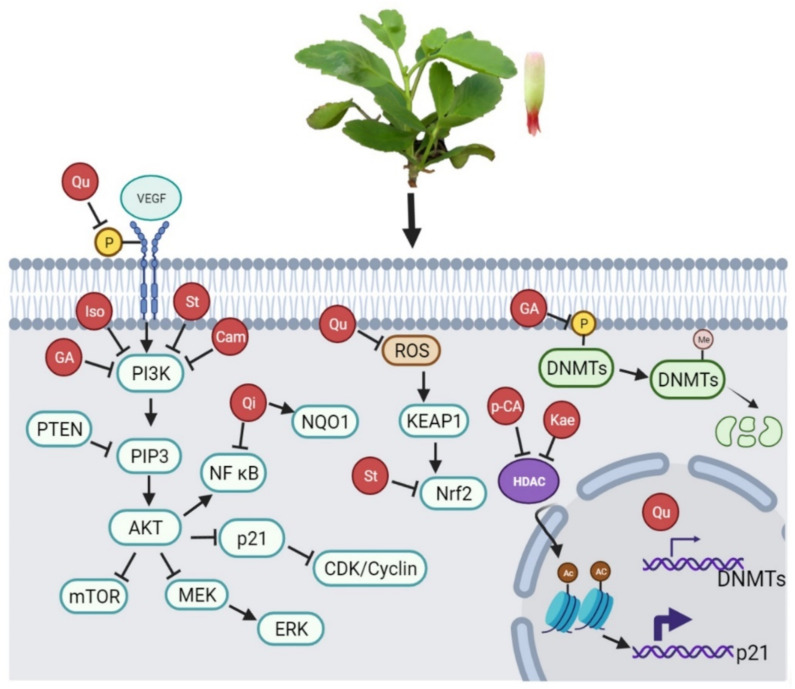
Pathways in which *K. pinnata* regulates gene expression. Phytochemicals of *K. pinnata* are involved in processes such as proliferation, apoptosis, cell migration, angiogenesis, metastasis, promoter methylation, inhibition of nuclear and cytoplasmic DNMTs, oxidative stress, and autophagy (see text for details). Source: Image created with BioRender.com.

**Table 1 molecules-27-06425-t001:** Anticancer properties found in the Kalanchoe genus.

Kalanchoe	Subject	Effect	Study Type	Cell Line	Reference
*K. daigremontiana* Raym.-Hamet and H. Perrier	Ovarian, cervical, breast cancer, and melanoma	Antiproliferative, cytotoxic, and antioxidant activity. Cell cycle arrest and caspase-independent cell death	In vitro	SCOV-3, HaCaT, HeLa, MCF-7, A375	[8,20,26,27,28]
*K. integra var. crenata* (Andr.)	Cardiotoxicity by doxorubicin Colorectal adenocarcinoma, lung cancer, mesothelioma, hepatocarcinoma, breast cancer	Cardio-protection against cardiotoxicity by cancer therapy Apoptosis	In vivo (rats) In vitro	DLD-1, A549, SPC212, HepG2, MCF-7	[9,29]
*K. tubiflora* (Harvey)	Lung cancer Lung adenocarcinoma, oral adenosquamous carcinoma, melanoma, and leukemia cell lines	Induction of autophagy Cell cycle arrest and senescence Cell cycle arrest and apoptosis Cytoprotective autophagy	In vitro In vitro, in vivo (mice)	CL1-5 A549, Cal-27, A2058, HL-60	[30,31,32]
*K. gastonis*-bonnieri Raym.-Hamet	Benign prostatic hyperplasia Prostate cancer	Antiproliferative activity and apoptosis induction Antiproliferative activity, apoptosis induction, and androgen receptor degradation	In vitro	Stromal cells LAPC-4, LNCaP, PC-3, DU145	[33,34]
*K. flammea*	Prostate cancer	Apoptosis induction and cell cycle arrest	In vitro	PC-3, LNCaP, PrEC	[35]
*K. laetivirens*	Lung cancer	Reversion of etoposide resistance	In vitro	A549, A549RT-eto	[36]
*K. gracilis* (L.) DC	Murine macrophage and human hepatocarcinoma	Antiproliferative, antioxidant, and anti-inflammatory activity	In vitro	RAW264.7, HepG2	[37]
*K. beharensis*	Acute myeloid leukemia	Apoptosis induction, inhibition of NF-κB	In vitro	HL-60, HL60R	[38]
*K. brasiliensis*	Kidney carcinoma	Cytotoxic activity	In vitro	3T3, 786-0	[39]
*K. laciniata*	Baby hamster kidney cell line	Cytotoxic activity	In vitro in vivo (mice)	BHK-21	[40]

## Data Availability

Not applicable.

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
