# Peer review of "Potential of Kalanchoe pinnata as a Cancer Treatment Adjuvant and an Epigenetic Regulator"

_molecules, 2022, doi:10.3390/molecules27196425_

Round 1

Reviewer 1 Report

The organization of the review is poor.

The sub-headings need redrafting to be more scientifically sound.

More figures should be added.

Author Response

Point 1: The organization of the review is poor.

Response 1: Dear reviewer, in response to your comment, we have modified the organization of the manuscript to give it more clarity, including a section called "4. Participation of K. pinnata in epigenetic regulation", which encompasses the epigenetic information found for the phytochemicals of K. pinnata and recently published information about its role in epitranscryptomic regulation.

Point 2: The sub-headings need redrafting to be more scientifically sound.

Response 2: We modified the sub-headings from some sections to give it more clarity. Modifications were marked in yellow.

Point 3: More figures should be added.

Response 3: We have added three figures more.

Additionally we make the following comment, the text had been revised in letpub.com, we sent it to a new revision at the English language editing by MDPI.

Reviewer 2 Report

Title: " Comparative Study on the Phenolic Fingerprint and Antioxidant Activity of Strawberry Tree (Arbutus unedo L.) Leaves and Fruits?"

Reviewer(s)' Comments to Author:

Referee:

Comments to the Author:

The authors have performed concise and comprehensive study on potential of Kalanchoe pinnata as a cancer treatment adjuvant 2 and an epigenetic regulator. So I suggest the authors go through some minor revisions to improve the quality of the manuscript.

1. Why authors selected phenolic compunds of Kalanchoe pinnata as a cancer treatment adjuvant 2 and an epigenetic regulator. Already phenolic compounds, flavnoids and others have been reported previously for cancer treatment. Authors should discuss the novelty of the present investigation.

2. Recent relevant lieratures regarding the characterization of polyphenols must be elaborate and discussed in the results and discussion section .

3. Some figures and tables shoould be included for improving the quality of manuscript.

4. Study is required a docking and molecular dynamics simulation study.

Author Response

Point 1:. Why authors selected phenolic compunds of Kalanchoe pinnata as a cancer treatment adjuvant 2 and an epigenetic regulator. Already phenolic compounds, flavnoids and others have been reported previously for cancer treatment. Authors should discuss the novelty of the present investigation.

Response 1: Dear reviewer, we did not make a selection aimed at the phenolic compounds of K. pinnata, the phytochemicals described and their function are the result of the information analysis obtained about the compounds it contains and the functions that have been found for them. We hope that the included section "4. Participation of K. pinnata in epigenetic regulation" will allow us to make the novelty of this research more clear.

Point 2:  Recent relevant lieratures regarding the characterization of polyphenols must be elaborate and discussed in the results and discussion section .

Response 2: We had included all the recent information available on the phytochemicals of K. pinata, now we point out which are the most studied polyphenols, indicating which has been analyzed in K. pinata

Point 3:  Some figures and tables shoould be included for improving the quality of manuscript.

Response 3: We have added more figures to give more clarity to the review.

Point 4: Study is required a docking and molecular dynamics simulation study.

Response 4:

Our review does not focus on computational strategies, basically, it is a bibliographic data integration, we know molecular docking and molecular dynamics are useful for elucidating the mechanism of drug action or design, but not considered for this work. We mentioned the result obtained by Du et al. 2022, an experimental work that uses a docking screening with quercetin. If you consider that we should include this analysis, we need more time to do it and restructure the manuscript.